# Can We Encode Intra- and Inter-Variability with Log Jacobian Maps Derived from Brain Morphological Deformations Using Pediatric MRI Scans?

**Andjela Dimitrijevic**[*1,2]                                        ANDJELA.DIMITRIJEVIC@POLYMTL.CA
[1] *NeuroPoly Lab, Institute of Biomedical Engineering, Polytechnique Montréal, Montréal, Quebec, Canada.*
[2] *Research Center, Ste-Justine Hospital University Centre, Montreal, Quebec, Canada.*

**Fanny Dégeilh**[*3]                                                  FANNY.DEGEILH@INSERM.FR
[3] *Univ Rennes, CNRS, Inria, Inserm, IRISA UMR 6074, EMPENN — ERL U-1228, Rennes, France.*

**Benjamin De Leener**[1,2,4]                                          BENJAMIN.DE-LEENER@POLYMTL.CA
[4] *Computer Engineering and Software Engineering, Polytechnique Montréal, Montreal, Quebec, Canada.*

## Abstract

Understanding individual variabilities in brain development is crucial for unraveling typical and atypical neurodevelopmental patterns. We propose a novel method using 3D CNN to characterize intra- and inter-individual variability based on log Jacobian maps derived from deformation fields. Inter pairs are chosen to match the distribution of intra pairs based on initial age, age interval (ia_r experiment) and also sex per pair (ias_r experiment). Training our model on log Jacobian maps, we explore two scenarios: one with overlaps between train and test sets derived from the same subjects, and the other with no overlaps, using 10-fold cross-validation. While both approaches achieved commendable results, the no overlap scenario showed slightly higher accuracy and F1 score. This research contributes to modeling neurodevelopmental trajectories for future deviation prediction.

**Keywords:** Registration, Deformation, Classification, Pediatric

## 1. Introduction

Understanding intra- and inter-individual variability in brain development is vital for characterizing typical neurodevelopment and detecting pathology (Bottenhorn et al., 2022). Inter-individual variability refers to differences between individuals (where minor differences are expected and significant ones may suggest a pathological condition), while intra-individual variability tracks changes over time within a person giving insights into brain development. During the early childhood developmental stage, inter variabilities are significant and can sometimes overshadow intra variabilities. This project aims to distinguish intra- and inter-individual variability using log Jacobian maps derived from deformable registration, focusing on pediatric brain development through longitudinal magnetic resonance imaging (MRI) scans. Recent studies have provided global growth curves for brain structures across the lifespan (Rutherford et al., 2022; Bethlehem et al., 2022). However, these do not capture intra-individual variabilities due to the cross-sectional nature of the

---

[*] Contributed equally

data. Distinguishing signatures for both intra- and inter-individual variabilities could help identify deviations from typical neurodevelopment. Our primary aim is to investigate the potential of log Jacobian maps in encoding intra and inter-variability. To achieve this, we propose a supervised method utilizing a 3D convolutional neural network (CNN).

## 2. Methods

Longitudinal T1-weighted MRI scans (N = 279 images) of 96 children (46 females) aged 2-to-7 years old from the Calgary Preschool Dataset (Reynolds et al., 2020) are used. 434 intra-individual (i.e., two images of the same child at two different ages; initial age: 3.93 ± 0.72 and age interval: 1.18 ± 0.03) pairs of images were first rigidly registered. Inter-individual pairs are chosen to either mirror intra pairs' distribution based on initial age and age interval (ia_r experiment, 433 pairs), or to also match the sex of each pair (ias_r experiment, 432 pairs). Affine registration was omitted to keep variations regarding the overall brain growth. Elastic SyN ANTs registration (Avants et al., 2008) was done to compute log Jacobian maps from the resulting deformations, normalizing images between -1 (local volume contraction) and 1 (local volume expansion), with zero indicating no change. The total number of pairs for both experiments (867 and 866) were separated into train (70%), validation (20%) and test (10%) sets. Two scenarios were examined: one with overlaps and one with no overlaps between image pairs in the train and test sets from the same subjects. While standard practice avoids data duplication in the test set, this study investigates if exposure to one time-point or underlying information from the same subject, via log Jacobian maps, aids in the classification task. The whole pipeline is available in Figure 1. Data augmentation is done by rotating volumes at different angles to enhance diversity. The architecture is a 17-layer 3D CNN inspired by Zunair et al. (2020) trained for 100 epochs and a batch size of 2 with a binary cross entropy loss. Lastly, label classification performance is evaluated with accuracy and F1 score (precision and recall) metrics.

## 3. Results & Discussion

Table 1 illustrates that the no overlap scenario consistently outperformed the overlap scenario in terms of average accuracy and F1 score across 10 splits for both experiments. Additionally, the experiment wherein inter pairs are matched to intra pairs based on initial age, age interval, and sex distribution yielded the highest accuracy and F1 score (0.989±0.019). Analysis of average absolute intra log Jacobians reveals a wider distribution with a higher mean, indicating larger local volume changes. In contrast, the distribution for inter pairs exhibits lower mean log Jacobian values and decreased variability, indicating attenuated local changes. This phenomenon may arise from local volume changes being overshadowed by more global deformations, as inter pairs typically involve larger expected differences compared to intra pairs. A visual representation of these distributions, categorized by age interval and pair type for both scenarios, is accessible in the GitHub repository (https://github.com/neuropoly/pediatric-intra-inter-encoding). It would be relevant to inspect log Jacobian values per segmented region to quantify differences between the intra and inter-individual variabilities.

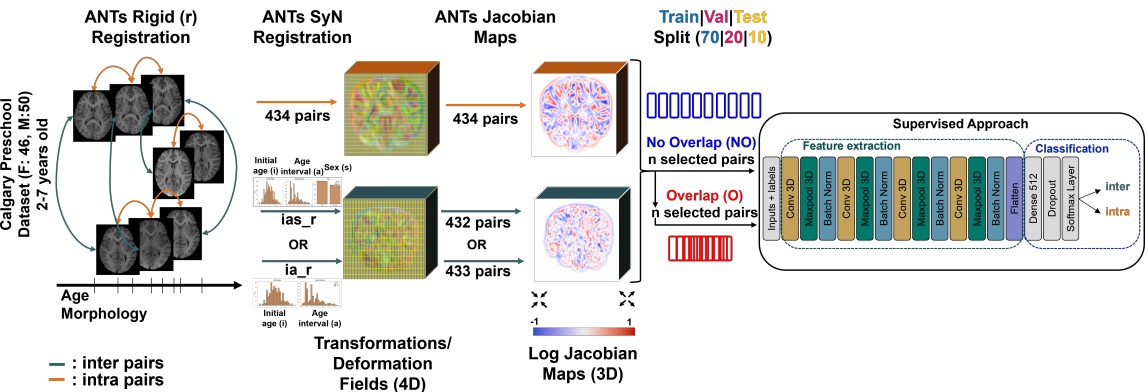

Figure 1: The pipeline involves rigid registration of intra and inter pairs from 96 subjects, followed by ANTs SyN registration. Jacobian maps are computed from the deformation and log-scaled. Ten sets with no overlaps yield 598 and 538 pairs for ia_r and ias_r experiments, respectively. In the overlap scenario, the same number of pairs were randomly selected whilst also ensuring a balanced 50/50 split between intra and inter pairs. Pairs are then fed into a 17-layer 3D CNN.

Table 1: F1 scores, accuracies and average absolute log Jacobian values for the two experiments with rigid initialization (ias_r and ia_r) and their scenarios with no overlap (NO) or overlap (O) in between sets. The number of total pairs is also shown.

| Experiment | F1 score (mean±std) | Accuracy (mean±std) | Avg abs log Jac intra/inter | Nbr total pairs (intra\|inter) |
|---|---|---|---|---|
| ias_r_NO | **0.989±0.019** | **0.989±0.019** | 0.029±0.006/ 0.025±0.002 | 538 (288\|250) |
| ias_r_O | 0.980±0.016 | 0.981±0.015 | 0.029±0.006/ 0.024±0.002 | 538 (269\|269) |
| ia_r_NO | 0.986±0.009 | 0.984±0.012 | 0.029±0.006/ 0.024±0.002 | 598 (338\|260) |
| ia_r_O | 0.984±0.014 | 0.985±0.014 | 0.029±0.006/ 0.024±0.002 | 598 (299\|299) |

## 4. Conclusion

The study effectively differentiates log Jacobian maps for intra and inter-individual variability in pediatric MRI, with both scenarios surpassing 0.98 accuracy and F1 scores. Leveraging the trained feature extraction module could aid in understanding brain development, aging, pathologies, and treatments by extracting neurodevelopmental trajectories. Future research involves exploring log Jacobian values in segmented regions and using calculated deformation mappings for trajectory prediction.

## Acknowledgments

This study was supported by Polytechnique Montréal, by the Canada First Research Excellence Fund, and by the TransMedTech Institute.

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
