# OpenReview forum: "Can We Encode Intra- and Inter-Variability with Log Jacobian Maps Derived from Brain Morphological Deformations Using Pediatric MRI Scans?"
_MIDL.io/2024/Short_Papers — MIDL 2024 Short Papers_

### Official Review · Reviewer_J2iC · 2024-04-24

**Confidence:** 3
**Final Rating:** 3.5

**Review:**

The authors perform an analysis on building a classifier out of log jacobian maps to determine whether the map is between or within subjects.

This seems like an obvious effect (That one should be easily able to predict this), but I don't recall any particular paper doing it in the past (though of course I may have missed it!) especially in Pediatric Scans.

The analysis and method seem pretty reasonable, with strong (albeit non-surprising) results. It can provide solid discussion at the conference among neuroscientists.

---

### Decision · Program_Chairs · 2024-04-26

Accept